

# Unlocking the potential of endothelial progenitor cells: a comprehensive review of definitions, applications, and future directions

Gongjie Ye[1,*], Yongfei Song[2,*], Yiru Weng[1], Jiangfang Lian[3], Jianqing Zhou[3] and Zhouzhou Dong[1]

[1] Department of Intensive Care Unit, Lihuili Hospital Affiliated of Ningbo University, Ningbo, Zhejiang, China
[2] Ningbo Institute of Innovation for Combined Medicine and Engineering, Lihuili Hospital Affiliated of Ningbo University, Ningbo, Zhejiang, China
[3] Department of Cardiovascular Medicine, Lihuili Hospital Affiliated of Ningbo University, Ningbo, Zhejiang, China
[*] These authors contributed equally to this work.

## ABSTRACT

Endothelial progenitor cells (EPCs) are undifferentiated cells with the capacity to mature into endothelial cells (ECs). EPCs have garnered considerable attention in the fields of regenerative medicine and cardiovascular therapy, owing to their pivotal role in neovascularization and vascular repair. Nonetheless, numerous challenges and questions persist regarding the translational research and practical application of EPCs. This review aims to examine the varying definitions of EPCs, their classification, extraction methods, and sources. It will also address the optimization of cultivation techniques for EPCs and the reprogramming of EPCs into induced pluripotent stem cells (iPSCs). Furthermore, the review will delve into the role of EPCs in cardiovascular diseases (CVD), septic shock, and rheumatic immune conditions, as well as their implications in connective tissue diseases (CTDs) and skin soft tissue regeneration. Finally, the article will discuss future research prospects for EPCs, aiming to engage and inspire readers.

## INTRODUCTION

The discovery of endothelial progenitor cells (EPCs) marked a significant shift in the understanding of vasculogenesis, which was previously thought to occur only during embryogenesis. EPCs were first isolated from human peripheral blood (PB) in 1997, challenging the traditional view and suggesting that these cells could contribute to the repair and regeneration of blood vessels in adults (*Asahara et al., 1997*). EPCs are derived from various sources, including bone marrow, spleen, and umbilical cord, and play a crucial role in the regeneration of the endothelial lining of blood vessels and wound repair. They are believed to originate from hematopoietic stem cells and mesenchymal stem cells,

Corresponding authors
Jianqing Zhou,
zhoujianqing5431@126.com
Zhouzhou Dong,
NBICUDONG@163.com

and their mobilization from bone marrow to peripheral circulation is highly regulated under both normal physiological conditions and stress (*Rana, Kumar & Sharma, 2018*). In addition to EPCs, ECs themselves are diverse and can be classified into different subtypes based on their location and function. Arterial and venous ECs, for instance, exhibit distinct phenotypic and functional characteristics. Arterial ECs are typically exposed to higher shear stress and have a more robust structure compared to venous ECs. The specification of ECs into arterial or venous subtypes is influenced by environmental cues, which can be leveraged to modulate pluripotent stem cell-derived endothelial cells (PSC-ECs) into a more homogeneous phenotype for clinical applications (*Arora, Yim & Toh, 2019*).

The understanding of EPCs and their interaction with different ECs subtypes continues to evolve, with ongoing research focused on elucidating their roles in vascular biology and potential therapeutic applications. Despite the initial enthusiasm, the field has faced challenges, particularly concerning the characterization and standardization of EPCs, as well as their clinical implications (*Resch et al., 2012*). The therapeutic applications of EPCs are also being explored in clinical settings. Strategies such as EPCs infusion therapy and the use of EPCs-capturing stents are being investigated to enhance endothelial repair and improve outcomes in patients with cardiovascular diseases (*Xiao & Kuang, 2021*; *Bianconi et al., 2018a*). However, challenges remain in standardizing EPCs isolation and characterization methods, as well as understanding the optimal conditions for their therapeutic use. Nonetheless, the study of EPCs and ECs remains a promising area of research with the potential to significantly impact cardiovascular therapy and regenerative medicine.

Finally, to enhance our understanding of EPCs, we employ a schematic diagram to visually depict their mechanism of action, angiogenesis processes, and associated signaling pathways (refer to Fig. 1 for further details).

## SEARCH METHODOLOGY SECTION

This review is specifically aimed at researchers with a focus on EPCs. We utilized a systematic search methodology for the literature review, which involved the formulation of specific research questions and the establishment of inclusion and exclusion criteria for selecting relevant studies. To ensure the rigor and impartiality of the review, we conducted an exhaustive examination of pertinent literature addressing the research questions. The review is structured into 11 sections, each of which will be independently searched and screened. Figure 2 and Table 1 outline the search strategy employed to identify relevant studies. The search process consisted of four distinct phases of search and refinement and was conducted across major online databases, including IEEE Xplore® (https://ieeexplore.ieee.org), ACM Digital Library, Google Scholar, SpringerLink, and Science Direct.

In Phase 1, the search string detailed in Table 1 was developed to extract pertinent studies from the aforementioned databases. This search string was formulated through an analysis of keywords identified in the relevant literature. Initially, the application of these search terms yielded a substantial number of potential studies within the databases. The

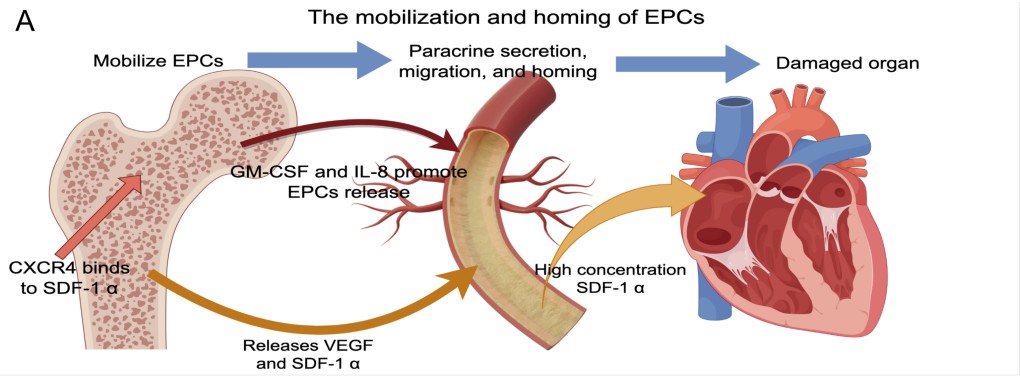

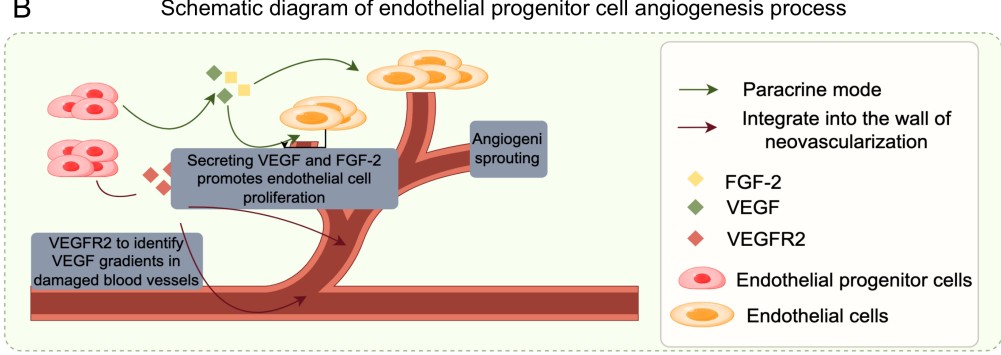

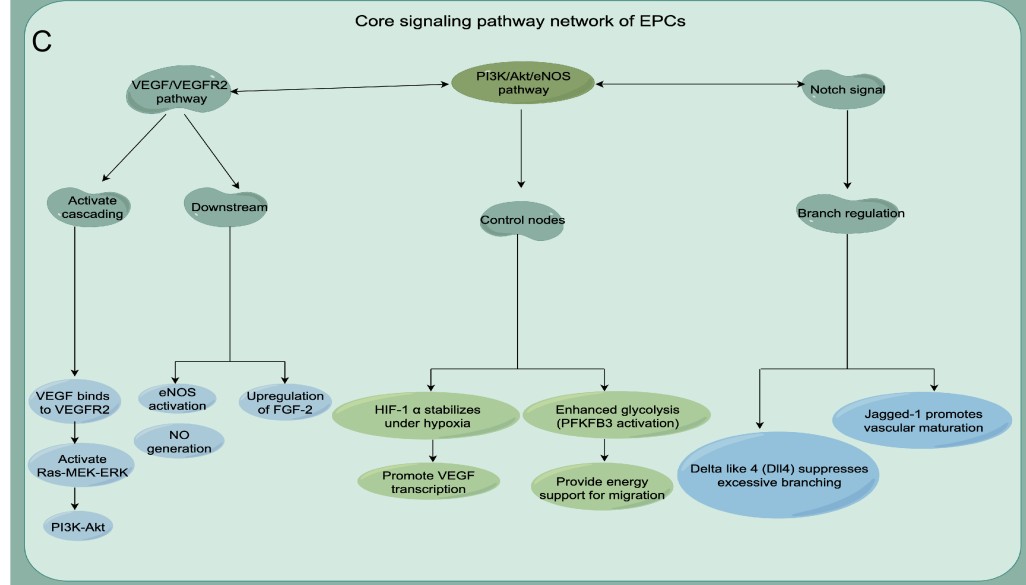

**Figure 1** **The schematic diagram to visually depict their mechanism of action, angiogenesis processes, and associated signaling pathways.** (A) Mobilization and homing of EPCs; (B) Schematic diagram of endothelial progenitor cell angiogenesis process; (C) The signaling pathway network of EPCs.

specific search strategy and the quantity of literature retrieved are presented in Table 1. In Phase 2, articles were excluded based on unavailability, redundancy, or lack of relevance. In Phase 3, inclusion and exclusion criteria were established to refine the filtering process

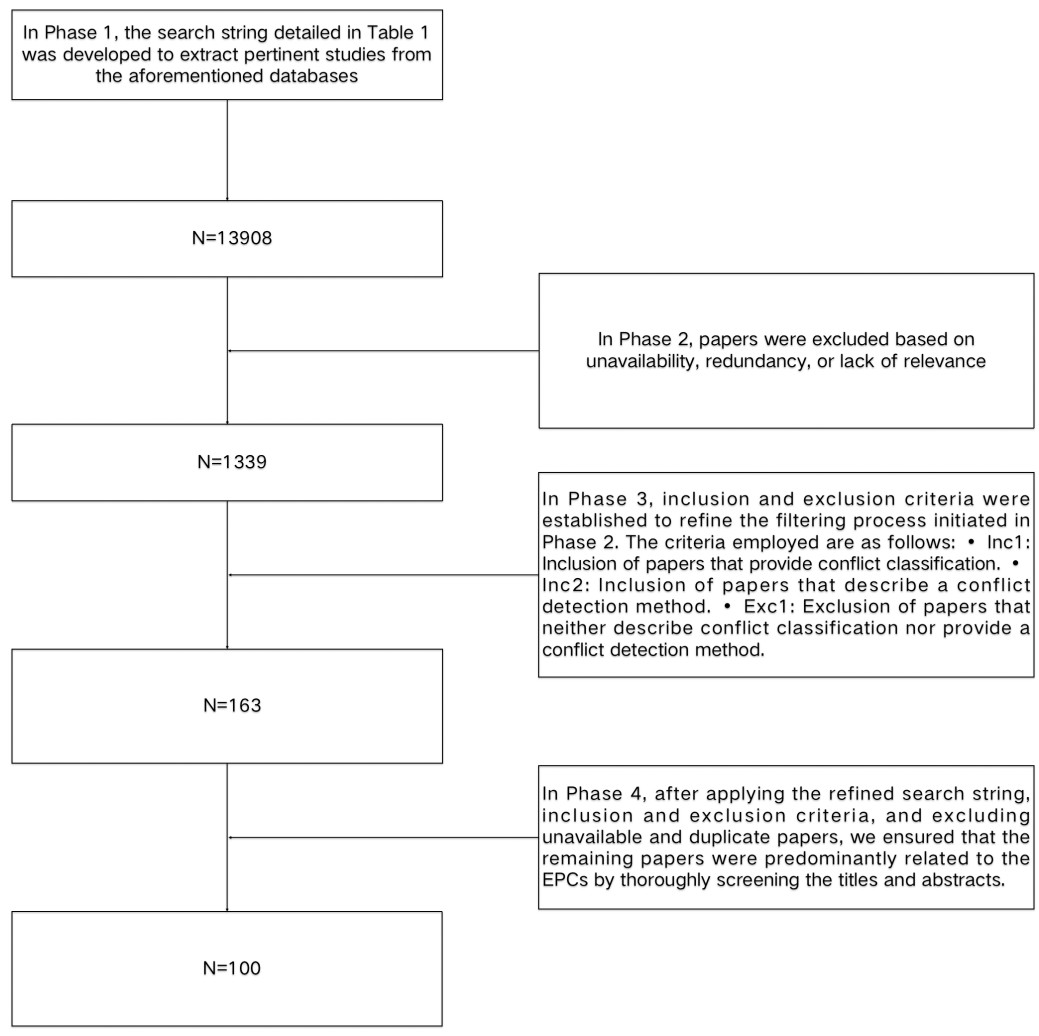

**Figure 2** A flowchart in the Search Methodology section.

initiated in Phase 2. The criteria employed are as follows: •Inc1: Inclusion of articles that provide conflict classification. •Inc2: Inclusion of articles that describe a conflict detection method. •Exc1: Exclusion of articles that neither describe conflict classification nor provide a conflict detection method. In Phase 4, after applying the refined search string, inclusion and exclusion criteria, and excluding unavailable and duplicate articles, we ensured that the remaining articles were predominantly related to the EPCs by thoroughly screening the titles and abstracts. Finally, we obtained 86 studies.

## Definition and classification of EPCs

EPCs are bone marrow-derived cells crucial for vascular repair and regeneration, circulating in the bloodstream to aid in vasculogenesis and endothelial repair (*Ozkok & Yildiz, 2018*; *Leszczynska et al., 2013*). The definition of EPCs remains debated, with researchers

Ye et al. (2025), *PeerJ*, DOI 10.7717/peerj.20128

**Table 1** The survey search methodology.

| | The search string In phase 1 | In phase 2 | In phase 3 | In phase 4 |
|---|---|---|---|---|
| 1. Divergence in the definition of EPCs | ("EPCs" or "endothelial progenitor cells") and "definition", $N = 200$ | $N = 93$ | $N = 29$ | $N = 23$ |
| 2. Early EPCs and late EPCs | ("early EPCs" or "late EPCs") and "category", $N = 700$ | $N = 270$ | $N = 20$ | $N = 9$ |
| 3. Extraction methods of EPCs | ("EPCs" or "endothelial progenitor cells") and "Extraction methods", $N = 109$ | $N = 46$ | $N = 8$ | $N = 5$ |
| 4. The sources of EPCs | ("EPCs" or "endothelial progenitor cells") and "source", $N = 7220$ | $N = 208$ | $N = 12$ | $N = 4$ |
| 5. Optimization of cultivation techniques for EPCs | ("EPCs" or "endothelial progenitor cells") and "cultivation", $N = 265$ | $N = 103$ | $N = 12$ | $N = 10$ |
| 6. EPCs and CVD | ("EPCs" or "endothelial progenitor cells") and "CVD", $N = 233$ | $N = 110$ | $N = 20$ | $N = 9$ |
| 7. EPCs and septic shock | ("EPCs" or "endothelial progenitor cells") and ("septic shock" or "sepsis"), $N = 96$ | $N = 49$ | $N = 9$ | $N = 8$ |
| 8. EPCs and rheumatic immunity, CTDs | ("EPCs" or "endothelial progenitor cells") and ("rheumatic immunity" or "CTD"), $N = 50$ | $N = 23$ | $N = 11$ | $N = 5$ |
| 9. EPCs and skin soft tissue regeneration | ("EPCs" or "endothelial progenitor cells") and ("skin" or "soft tissue") and "regeneration", $N = 4960$ | $N = 380$ | $N = 22$ | $N = 8$ |
| 10. Reprogramming of EPCs into iPSCs | ("EPCs" or "endothelial progenitor cells") and "iPSCs" and "reprogramming", $N = 52$ | $N = 40$ | $N = 10$ | $N = 11$ |
| 11. Research prospect of EPCs | ("EPCs" or "endothelial progenitor cells") and "Research prospect", $N = 23$ | $N = 17$ | $N = 10$ | $N = 8$ |

**Notes.**
EPCs, endothelial progenitor cells; iPSCs, induced pluripotent stem cells; CVD, cardiovascular disease; CTDs, connective tissue diseases.

advocating for immunophenotyping and functional assays to distinguish different EPCs subtypes (*Markeson et al., 2015*).

Several researchers propose that cells expressing CD34, CD133, and KDR markers, known as $CD34^+CD133^+KDR^+$ triple-positive cells, are categorized as EPCs (*Arica et al., 2019*; *Cesari et al., 2008*). These cells have attracted considerable interest due to their potential to differentiate into mature endothelial cells. Moreover, the functional properties of $CD34^+CD133^+KDR^+$ EPCs are closely linked to their clonogenic potential and angiogenic capabilities. Studies have shown that these cells exhibit high clonogenicity, which is a predictor of their ability to proliferate and form new blood vessels. The initial clonogenic potential of these cells is indicative of their future functionality, suggesting that selecting for high-quality progenitor cells can enhance therapeutic outcomes in ischemic conditions (*Ferratge et al., 2017*; *Atashi et al., 2018*).

Researchers are increasingly focusing on EPCs subpopulations with lineage-negative markers like $CD14^-/CD45^-$, which may enhance angiogenesis. Studies indicate that EPCs function differently under various pathological conditions. For instance, in emphysema, $CD45^-/CD31^+/CD34^+$ EPCs aid lung endothelial regeneration and angiogenesis (*Pakhomova et al., 2020*). Similarly, in chronic obstructive pulmonary disease (COPD), the drug Spikerone boosts pulmonary microcirculation regeneration by mobilizing $CD45^-/CD34^+/CD31^+$ EPCs (*Skurikhin et al., 2021*). Research indicates that the $CD14^-$ subset can form EPCs colonies only when co-cultured with the $CD14^+$ subset, implying that $CD14^-$ may be the source of EPCs. This colony formation relies on cytokines from the CD14 subset, especially angiopoietin 1, offering new insights into EPCs origins and functions (*Sudchada et al., 2012*). These findings highlight the therapeutic potential of lineage-negative EPCs across different pathological contexts.

Certain studies contend that EPCs have the ability to differentiate into mature endothelial cells exhibiting a "cobble-stone" morphology (*Ni et al., 2019*; *Badawi et al., 2022*). Previous study has corroborated this finding (Fig. 3) (*Ye et al., 2014*). Recent studies have highlighted the importance of the microenvironment in promoting the differentiation of EPCs. For instance, the presence of specific growth factors and extracellular matrix components can significantly enhance the maturation of these cells into functional endothelial cells. The differentiation process transforms these progenitor cells into what can be described as "cobble-stone" in the vascular landscape, contributing to the integrity and functionality of blood vessels. This maturation is not merely a change in morphology but also involves a complex reprogramming of gene expression that equips these cells with the necessary capabilities to participate in angiogenesis and vascular repair (*Plein et al., 2018*; *Kanaya et al., 2015*).

Studies indicate that EPCs can be identified by their uptake of acetylated low density lipoprotein (ac-LDL) and binding to Ulex europaeus agglutinin-1 (UEA-1) (*Ye et al., 2014*; *Wang et al., 2021*; *He et al., 2014*). LDL is vital in cardiovascular health and endothelial function, affecting EPCs, which are important for vascular repair. The interaction between LDL and EPCs, particularly ac-LDL uptake and UEA binding, is crucial for EPCs differentiation and function. EPCs that take up ac-LDL show improved capillary

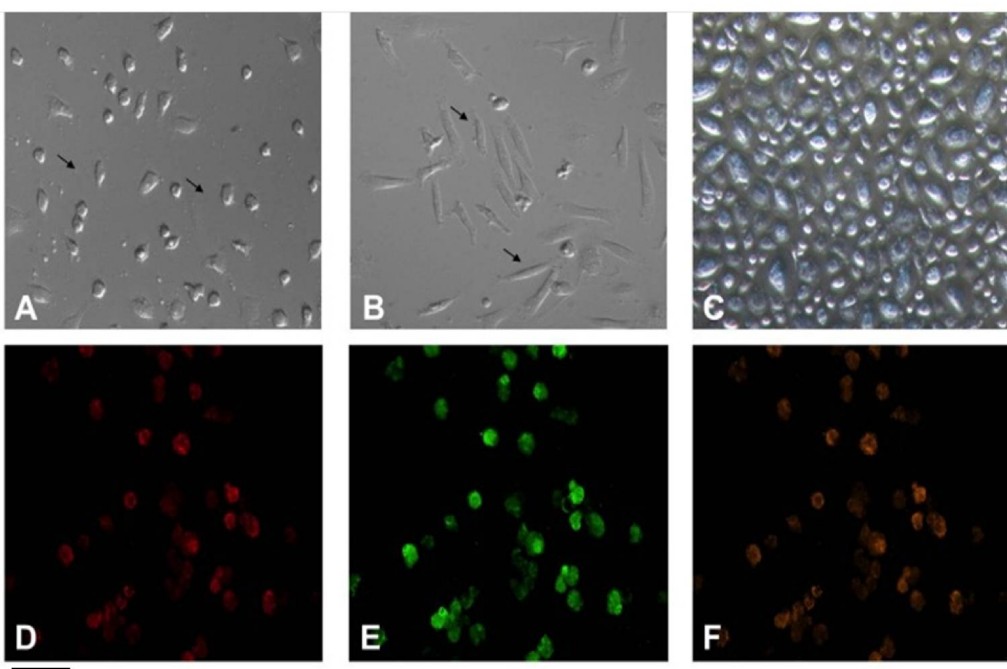

**Figure 3 Characterization of EPCs derived from human peripheral blood.** (A) The ellipsoid morphology of EPCs derived from human PB on day seven of culture. (B) The spindle-shaped morphology of EPCs derived from human PB on day14 of culture. (C) The "cobble-stone" morphology of EPCs derived from human PB on day 21 of culture; (D) fluorescence image shows uptake of DiI-Ac-LDL; (E) fluorescence image demonstrate expression of FITC-UEA lectin in EPCs; (F) all DiI-labeled acetylated LDL(+) cells stained positive for FITC-ulex-lectin binding, as can be seen in overlay. Scale bar: A–F = 100 μm. Adapted from *Ye et al. (2014)*.

formation, essential for repairing damaged endothelium and restoring vascular function (*Chen et al., 2011*; *Hong et al., 2020*).

Moreover, the binding of UEA to EPCs serves as a marker for identifying these cells and assessing their functionality. UEA binds to specific carbohydrate structures on the surface of endothelial cells, which can be indicative of the cells' maturity and their ability to participate in angiogenesis. The interaction between UEA and EPCs can provide insights into the cellular mechanisms that govern endothelial repair processes, particularly in the context of oxidative stress and inflammation induced by oxidized LDL (ox-LDL) (*Liu, Gao & Wang, 2023*; *Wang et al., 2020*). In addition to their role in vascular repair, the relationship between LDL and EPCs has implications for understanding the pathophysiology of cardiovascular diseases. Elevated levels of LDL, particularly in its oxidized form, are associated with endothelial dysfunction and increased cardiovascular risk. Studies have shown that ox-LDL can impair EPCs function, leading to reduced angiogenic potential and contributing to the progression of atherosclerosis (*Wang et al., 2020*; *Zenti & Stefanutti, 2011*). Therefore, understanding the dynamics of LDL and EPCs interactions is crucial for developing therapeutic strategies aimed at enhancing endothelial repair and mitigating cardiovascular disease risk.

In conclusion, EPCs may not represent a singular cell type; instead, they constitute a heterogeneous group of cells sharing the capacity to differentiate into endothelial cells with angiogenic properties. Classical EPCs are characterized by three primary features: (1) the expression of CD34, CD133, and KDR markers, with the gradual loss of CD133 as the cells mature; (2) the ability to uptake ac-LDL and bind to UEA-I; and (3) the potential to differentiate into mature endothelial cells, which exhibit a characteristic "cobble-stone" morphology.

### Early EPCs vs. Late EPCs

EPCs are classified into two main categories: early EPCs and late EPCs, each playing distinct roles in vascular repair and regeneration. Cells identified as $CD34^+KDR^+$ are categorized as late EPCs, while those characterized by the markers $CD34^+CD133^+KDR^+$ are considered early EPCs, as delineated in previous research (*Arica et al., 2019*).

Early EPCs initiate endothelial repair by secreting angiogenic factors and promoting neovascularization through paracrine mechanisms. In contrast, late EPCs excel in proliferation, migration, and differentiation into mature endothelial cells, vital for ongoing vascular repair and maintaining endothelial integrity (*Cheng et al., 2013*; *Tagawa et al., 2015*). These functional differences are reflected in their gene expression: early EPCs express more inflammatory cytokines and paracrine factors to recruit cells to injury sites, while late EPCs have genes linked to proliferation and angiogenesis, aiding in endothelial tubulogenesis and neovascularization (*Ke et al., 2017*; *Li et al., 2012*). This distinction is crucial for vascular repair, especially in conditions like coronary artery disease and ischemic injuries. Research indicates that late EPCs outperform early EPCs in functionality, exhibiting higher proliferation and enhanced tube formation crucial for angiogenesis. These cells express key endothelial markers and aid in blood vessel repair, making them significant in cardiovascular research and therapy (*Fernandez et al., 2014*; *Paschalaki & Randi, 2018*). The differentiation of EPCs into mature endothelial cells is complex, involving multiple factors. Understanding the differences between early and late EPCs and the conditions promoting their maturation is vital for their clinical use, especially in cardiovascular disease and tissue engineering (*Kumboyono et al., 2021*; *Ablin et al., 2011*). Table 2 presents a comparative analysis of the similarities and differences between early EPCs and late EPCs.

In summary, the roles of early and late EPCs in vascular biology are complementary, with early EPCs initiating the repair process and late EPCs ensuring its completion and maintenance.

## Sources and extraction methods of EPCs
### Various sources of EPCs

The isolation of EPCs can be achieved from multiple sources, including PB, umbilical cord blood (UCB), and bone marrow. Each of these sources presents unique advantages and challenges that can influence the yield and functionality of the isolated EPCs.

PB is often considered a convenient source for EPCs isolation due to its accessibility and the relatively non-invasive nature of collection. Studies have demonstrated that EPCs can be effectively isolated from PB mononuclear cells (PBMCs) using techniques such as

**Table 2 Comparative analysis of early EPCs *vs.* late EPCs.**

| Feature | Early EPCs | Late EPCs |
|---|---|---|
| Source | Peripheral blood, bone marrow, monocytes/macrophages | Peripheral blood, bone marrow, umbilical cord blood, vascular endothelia |
| Culture time | Emerge within 4–7 days | Appear after 2–4 weeks |
| Morphology | Spindle-shaped | Cobblestone-shaped |
| Proliferation capacity | Low proliferative potential | High proliferative potential |
| Surface Markers | $CD34^+$, $CD133^+$, $VEGFR\text{-}2^+$, $CD31^+$ | $CD34^-$, $CD133^-$, $VEGFR\text{-}2^{++}$, $VE\text{-}cadherin^{++}$, $vWF^{++}$, $CD146^+$ |
| Functional roles | Secrete angiogenic cytokines (VEGF, IL-8), Modulate inflammation | Form vascular networks, Produce NO, Integrate into endothelial layers |
| Lifespan | Short (3–4 weeks) | Long (up to 12 weeks) |
| Oxidative stress | Susceptible to oxidative damage | Resistant due to higher eNOS and SOD activity |
| Therapeutic use | Cytokine secretion for paracrine effects | Neovascularization via direct endothelial integration |
| Clinical relevance | Associated with acute vascular repair | Linked to chronic vascular remodeling |

**Notes.**
Table 2 presents a comparative analysis of the similarities and differences between early EPCs and late EPCs.

density gradient centrifugation. However, the overall number of EPCs obtained from PB may be limited compared to other sources. Studies have shown that the culture conditions, such as the type of substrate and the presence of growth factors like vascular endothelial growth factor (VEGF), can significantly affect the yield and functionality of the EPCs obtained. For instance, a study demonstrated that culturing PBMCs on fibronectin in the presence of high VEGF concentrations resulted in improved EPCs proliferation and differentiation compared to other substrates and conditions (*Wu et al., 2012*).

UCB is another promising source for EPCs. It is rich in hematopoietic stem cells and has been shown to contain a substantial population of EPCs. The advantages of using UCB include its availability and the ethical considerations surrounding its collection, as it is obtained after childbirth without harm to the donor. Research indicates that UCB-derived EPCs exhibit robust proliferative capacity and can effectively contribute to vascular repair (*Phuc et al., 2012*). Furthermore, studies have highlighted the potential of UCB to yield multiple types of stem cells, including hematopoietic stem cells (HSCs) and mesenchymal stem cells (MSCs), which can be beneficial for various regenerative therapies (*Bhartiya et al., 2012*).

Bone marrow remains a traditional source for EPCs isolation, particularly in clinical settings. Bone marrow-derived EPCs have been extensively studied for their role in neovascularization and tissue repair. The isolation process typically involves aspiration and subsequent processing to obtain mononuclear cells, which can then be cultured to enrich for EPCs. While bone marrow-derived EPCs have demonstrated significant regenerative potential, the invasive nature of bone marrow collection and the associated risks can limit its use in some patient populations (*Jin et al., 2014*).

In summary, while PB, UCB, and bone marrow each serve as viable sources for EPCs isolation, the choice of source may depend on specific clinical needs, the desired quantity of cells, and the associated ethical considerations.

### Extraction methods

EPCs are vital for vascular repair and regeneration, and isolating them is key for therapeutic uses. The immunomagnetic bead method, which uses antibodies on magnetic beads to capture EPCs, ensures high purity and viability, making it popular in regenerative medicine. Another common technique is density gradient centrifugation, which separates cells based on density. This method effectively isolates PBMCs for further EPCs extraction, enhancing cell recovery and reducing contamination (*Papadimitriou et al., 1996*; *Belkhir et al., 2016*). Combining immunomagnetic bead methods with density gradient centrifugation improves EPCs isolation efficiency. Initially, density gradient centrifugation enriches PBMCs, followed by immunomagnetic separation to specifically isolate EPCs. This two-step process enhances EPCs purity and functionality, crucial for cell therapy and tissue engineering applications (*Fu et al., 2017*).

Fluorescence-activated cell sorting (FACS) is a crucial technique in cell biology for precisely sorting and analyzing cell types, including EPCs, based on specific markers. It is invaluable for studying EPCs and has diverse applications, such as isolating circulating tumor cells (CTCs) with minimal contamination by combining immunomagnetic enrichment with FACS. This method highlights FACS's versatility in isolating rare cell populations for molecular analyses (*Say et al., 2013*). Additionally, FACS is used to study dendritic cell subsets in human atherosclerotic plaques, emphasizing its role in identifying immune cells in complex samples (*Magbanua & Park, 2013*).

### Optimization of EPCs cultivation techniques

The cultivation of EPCs represents a pivotal area of research. Numerous studies have investigated diverse methodologies to enhance the proliferation and functionality of EPCs, emphasizing the significance of scaffold materials, growth factors, and culture conditions. These cultivation techniques are applicable across all EPCs subtypes, including endothelial colony-forming cells (ECFCs), early and late EPCs.

One significant advancement in EPCs cultivation is the use of specific scaffolds that enhance cell adhesion and proliferation. For instance, a study demonstrated that EPCs could be effectively cultivated on $\beta$-tricalcium phosphate ($\beta$-TCP) granules without the need for fibronectin coating, which is traditionally used to improve cell attachment. This research indicated that the design of the scaffold significantly influences cell behavior, with structural differences affecting both adherence and metabolic activity of EPCs (*Störmann et al., 2019*). Additionally, the incorporation of growth factors such as erythropoietin and granulocyte-monocyte colony-stimulating factor has been shown to further enhance the proliferation and differentiation of EPCs when cultured on fibrin scaffolds (*Grieb et al., 2011*).

The choice of culture media plays a pivotal role in the success of EPCs cultivation. A comparative analysis of various endothelial cell culture media revealed that defined media containing specific growth factors like EGF, FGF2, and VEGF significantly improved the outgrowth and viability of endothelial cells derived from both UCB and PB (*Ye et al., 2014*; *Leopold et al., 2019*). This finding underscores the necessity of selecting appropriate culture conditions to maximize the yield and functionality of EPCs. Another innovative approach involves the use of human platelet lysate as a substitute for animal serum in the culture of

ECFCs. This method not only adheres to good manufacturing practices but also enhances cell viability and proliferation rates, making it a promising technique for clinical applications (*Denecke et al., 2015*). Long-term culture of endothelial progenitor-like cells from adipose-derived stem cells in endothelial growth media can cause significant morphological and functional changes, suggesting their potential in vascular repair (*Amerion et al., 2018*). Recent studies have explored the molecular mechanisms of EPCs function, showing that inhibiting glycogen synthase kinase $3\beta$ boosts EPCs proliferation and migration in hypercholesterolemic conditions, and PDGFR-$\beta$ phosphorylation is linked to their angiogenic potential. These findings emphasize the role of signaling pathways in enhancing EPCs therapeutic efficacy (*Cui et al., 2015*; *Lu et al., 2017*).

Purifying high-purity EPCs is essential for effective research and therapy. Methods like FACS and two-dimensional preparative chromatography have shown promise. FACS isolates specific cell populations using fluorescence, while chromatography, using novel polar copolymerized RP stationary phases, achieves high purity and recovery from complex mixtures. Both techniques offer advantages for EPCs purification (*Larcher et al., 2018*; *Jin et al., 2013*). The development of a high-resolution purification method for CD34-negative severe combined immune deficiency (SCID)-repopulating cells from human cord blood, using monoclonal antibodies and fluorescence-activated cell sorting, offers valuable techniques for EPCs purification (*Ishii et al., 2011*). These methods, initially designed for other cells, can be adapted to improve the purity and consistency of EPCs for research and clinical use.

Overall, the cultivation techniques for EPCs are rapidly evolving, with ongoing research aimed at optimizing conditions to enhance their regenerative capabilities. The integration of advanced scaffolding materials, growth factor supplementation, and molecular targeting strategies holds promise for improving the efficacy of EPCs in clinical settings, particularly in the treatment of ischemic diseases and tissue regeneration.

## EPCs in regenerative medicine and disease treatment

The use of EPCs in clinical settings involves several key conditions and considerations: source of EPCs, characterization and purification, clinical Indications, delivery methods, safety and efficacy. Understanding EPCs biology is essential, as they aid in re-endothelialization and neovascularization, crucial for tissue repair and vascular health. Despite interest, progress is hindered by EPCs population heterogeneity and the absence of standardized isolation and expansion methods (*Wang et al., 2013*; *Keighron et al., 2018*). Identifying specific EPCs subtypes, like ECFCs, which have shown positive effects in preclinical studies, is crucial (*Keighron et al., 2018*). A key factor is optimizing EPCs delivery and function. Strategies like genetic modifications and co-culturing with other cells, such as bone marrow-derived mesenchymal stem cells (BMSCs), can enhance EPCs efficacy by promoting angiogenesis and improving outcomes in conditions like intrauterine adhesion (*Yu et al., 2018*). However, clinical application faces challenges such as cell survival, integration, and risks of adverse effects. Innovative tissue-engineered carrier matrices may help overcome these issues by supporting EPCs survival and growth (*Balaji et al., 2013*).

### EPCs and cardiovascular disease

EPCs have emerged as critical players in the context of cardiovascular disease (CVD), particularly due to their role in vascular repair and regeneration. These cells, which are derived from the bone marrow, are mobilized into the bloodstream in response to vascular injury and are essential for maintaining endothelial integrity. Their functionality and numbers are often compromised in patients with various cardiovascular risk factors, leading to impaired vascular repair mechanisms and contributing to the progression of CVD (*Madonna, Novo & Balistreri, 2016*; *Lorenzen et al., 2010*).

Circulating endothelial progenitor cells (cEPCs) are a type of undifferentiated cells derived from bone marrow or vascular walls, with the potential to differentiate into mature endothelial cells and migrate to PB under physiological or pathological stimuli, participating in angiogenesis and injury repair. Research indicates that fewer cEPCs increase cardiovascular event risks like heart attacks and strokes (*Lorenzen et al., 2010*). Studies show that patients with chronic kidney disease and plaque psoriasis have lower EPCs levels linked to higher cardiovascular risks, influenced by inflammation and endothelial dysfunction (*Michalska et al., 2020*). These conditions impair EPCs function *via* pathways like NF-$\kappa$B, NLRP3, and p38 MAPK. However, small molecule inhibitors and activators can restore EPCs activity by managing oxidative stress, inflammation, and metabolic reprogramming. Future research should focus on targeted delivery and multi-target strategies to boost EPCs' therapeutic potential in vascular repair.

The therapeutic potential of EPCs has been widely studied, with human embryonic stem cells (hESCs) being differentiated into EPCs that are vital for vascular repair. Generating EPCs with specific phenotypes is crucial for cardiovascular therapies, as they aid in re-endothelialization and neo-vascularization. EPCs from hPSCs express functional receptors like TLR4, which boosts their proliferation and maintains their stem cell characteristics, increasing their availability for therapeutic applications (*He et al., 2010*). Using hPSCs enables the study of new signaling pathways and molecular mechanisms to boost EPCs functionality. Activating pathways like AKT can enhance EPCs migration and tube formation, crucial for vascular repair. This understanding may lead to targeted therapies that enhance EPCs-based treatments' effectiveness (*Hu et al., 2015*). Furthermore, the role of EPCs as biomarkers for cardiovascular risk has gained attention. Studies have indicated that low levels of circulating EPCs are predictive of adverse cardiovascular outcomes, including mortality (*Rigato, Avogaro & Fadini, 2016*). Previous study found that decreased EPCs levels, particularly within the CD34$^+$/CD133$^+$/KDR$^+$ cell subsets, were linked to higher mortality rates in patients experiencing acute myocardial infarction (AMI) (Table 3) (*Ye et al., 2023*). This underscores the potential of using EPCs counts as a clinical tool for assessing cardiovascular risk and guiding therapeutic strategies.

In conclusion, EPCs play a pivotal role in cardiovascular health, serving both as a marker of vascular integrity and a potential therapeutic target. Their involvement in the pathophysiology of cardiovascular disease emphasizes the need for further research to fully understand their mechanisms and to develop effective strategies for enhancing their function in clinical settings (*Ye et al., 2023*; *Balistreri et al., 2015*; *Chiva-Blanch et al., 2014*).

**Table 3  Comparison of EPCs between survival group and death group in AMI patients ($M \pm SD$).**

| The category | Survival group ($N = 68$) | Death group ($N = 23$) | $t/U/x^2$ | $P$ |
|---|---|---|---|---|
| $CD34^+/CD133^+$ cells (%) | $0.50 \pm 0.17$ | $0.37 \pm 0.18$ | 19.56 | 0.00 |
| $CD34^+/CD133^+/KDR^+$ EPCs (%) | $0.19 \pm 0.06$ | $0.14 \pm 0.02$ | 28.96 | 0.00 |

Notes.

A statistically significant increases counts $CD34^+/CD133^+/KDR^+$ EPCs were observed in the survival group compared with the death group in AMI patients ($P < 0.05$). Adapted from *Ye et al. (2023)*.

### EPCs and septic shock

EPCs are crucial in septic shock for vascular repair and maintaining endothelial integrity. Sepsis triggers systemic inflammation, causing endothelial dysfunction and leading to multiple organ dysfunction syndrome (MODS) with high mortality rates. The correlation between organ failures and mortality is crucial for assessing patient outcomes. Research indicates that mortality risk escalates with more organ failures. For example, a study on critically ill patients revealed a 25% overall mortality rate, which rose sharply with each additional organ failure, underscoring the severe effect of multiple organ dysfunctions on survival (*Yasumoto et al., 1994*). EPCs mobilization and function are vital for reducing sepsis effects and aiding recovery from endothelial damage. Recent studies have highlighted the relationship between circulating EPCs levels and the severity of septic shock. For instance, research indicates that patients with septic shock exhibit altered levels of circulating EPCs, which are associated with the clinical course of the disease. Specifically, higher levels of circulating EPCs have been observed in patients with less severe forms of sepsis, while those with more severe septic shock show diminished EPCs mobilization and function (*Krautkrämer et al., 2014*; *Liu et al., 2018*). This suggests that the ability of EPCs to respond to endothelial injury is compromised in severe cases, potentially exacerbating the condition. The therapeutic potential of EPCs in septic shock has been explored through various interventions aimed at enhancing their mobilization and function. For example, the administration of specific growth factors has been shown to increase EPCs levels in circulation, which may aid in re-endothelialization and restoration of vascular integrity (*De Biasi et al., 2015*; *Edwards et al., 2018*). Additionally, the paracrine effects of EPCs, mediated through the release of microvesicles containing pro-angiogenic factors, have been implicated in promoting endothelial repair and reducing inflammation during septic episodes (*Zhang, Malik & Rehman, 2014*; *Hubert et al., 2014*).

The interplay between EPCs and the inflammatory milieu in septic shock is complex. Neutrophils, which are often activated during sepsis, can influence EPCs behavior by enhancing their angiogenic properties and promoting their recruitment to sites of injury. However, excessive inflammation may also hinder EPCs function, leading to impaired endothelial repair mechanisms. This dual role of EPCs as both protectors and potential victims of the inflammatory response underscores the need for targeted therapeutic strategies that can enhance their beneficial effects while mitigating the adverse impacts of sepsis-induced inflammation (*Li et al., 2012*; *Hubert et al., 2014*).

In summary, EPCs play a pivotal role in the pathophysiology of septic shock, with their levels and functionality serving as important indicators of disease severity and recovery potential.

### EPCs in rheumatic immunity and connective tissue diseases

The relationship between EPCs and rheumatic immunity, particularly in the context of connective tissue diseases (CTDs), is an area of growing interest in the field of immunology and rheumatology. EPCs are crucial for the maintenance and repair of the endothelium, and their dysfunction has been implicated in various autoimmune conditions, including systemic lupus erythematosus (SLE) and systemic sclerosis (SSc). In patients with these diseases, an imbalance between endothelial injury and repair mechanisms can lead to significant cardiovascular complications, which are prevalent in this population. In a study focusing on polymyalgia rheumatica (PMR), researchers found that patients exhibited a marked increase in circulating endothelial microparticles (EMPs) and a decrease in EPCs, indicating a state of endothelial dysfunction. This imbalance was closely associated with systemic inflammation, as evidenced by elevated C-reactive protein (CRP) levels. Notably, treatment with corticosteroids resulted in a significant reduction of both CRP and the EMP/EPCs ratio, suggesting that controlling inflammation can restore some degree of endothelial repair capacity in these patients (*Pirro et al., 2012*).

Moreover, the role of EPCs extends beyond mere repair; they are also involved in the modulation of immune responses. In the context of autoimmune diseases, the dysregulation of EPCs function may contribute to the chronic inflammatory state observed in conditions like rheumatoid arthritis (RA) and SLE. The presence of autoantibodies and inflammatory cytokines can adversely affect EPCs mobilization and function, leading to impaired angiogenesis and further exacerbating vascular complications associated with these diseases (*Arida et al., 2018*; *Zanatta et al., 2019*). Research has also indicated that the therapeutic targeting of EPCs could represent a novel approach in managing cardiovascular risks in patients with rheumatic diseases. For instance, interventions aimed at enhancing EPCs mobilization or function may help mitigate the vascular damage that often accompanies chronic inflammation in CTDs. This is particularly relevant given the established link between inflammation and accelerated atherosclerosis in these patients, which underscores the need for integrated management strategies that address both autoimmune and cardiovascular aspects of care (*Chen et al., 2021*; *Santos-Moreno et al., 2021*).

In conclusion, the interplay between EPCs and rheumatic immunity highlights a critical area of research that could lead to improved therapeutic strategies for patients with CTDs. Understanding the mechanisms that govern EPCs function in the context of autoimmunity will be essential for developing targeted interventions that can enhance endothelial repair and reduce cardiovascular morbidity in this vulnerable population.

### EPCs in skin and soft tissue regeneration

EPCs play a crucial role in the regeneration of skin soft tissues, particularly in the context of wound healing and tissue engineering. These cells are essential for the formation of new blood vessels, a process known as angiogenesis, which is vital for supplying nutrients and oxygen to regenerating tissues. Recent studies have highlighted the significance of EPCs

in enhancing the vascularization of engineered skin substitutes, thereby improving their integration and functionality when applied to full-thickness skin defects. For instance, the incorporation of EPCs into dermal scaffolds has been shown to significantly increase microvessel density and promote collagen synthesis, which are critical factors for effective skin regeneration (*Auxenfans et al., 2012*; *Meruane, Rojas & Marcelain, 2012*). In the realm of ear reconstruction, the application of EPCs has shown promising results. The regeneration of auricular structures requires not only the restoration of skin but also the re-establishment of vascular networks to support the newly formed tissue. Research indicates that EPCs can enhance the vascularization of auricular scaffolds, thereby improving the overall success of ear reconstruction procedures. By promoting angiogenesis, EPCs facilitate the integration of engineered tissues with the host vasculature, which is essential for the survival and functionality of the reconstructed ear (*Frueh et al., 2017*; *Otto et al., 2022*).

The combination of EPCs with other cell types, such as adipose-derived stem cells (ASCs) and fibroblasts, has been explored to further enhance tissue regeneration. This synergistic approach not only improves the vascular supply but also supports the structural integrity and functional recovery of the engineered tissues. For example, studies have demonstrated that the co-culture of EPCs with ASCs leads to improved outcomes in skin regeneration, as these cells work together to create a more favorable microenvironment for healing (*Patschan et al., 2016*; *Jeon, Joo & Cha, 2020*). The potential of using EPCs in tissue engineering extends beyond skin and ear reconstruction. Their ability to promote angiogenesis and support tissue regeneration makes them a valuable component in various regenerative medicine applications. As research continues to uncover the mechanisms by which EPCs contribute to tissue healing, their incorporation into engineered constructs is likely to become a standard practice in the field of regenerative medicine (*Cheung et al., 2014*; *Goyer et al., 2019*).

In conclusion, EPCs are pivotal in the regeneration of skin soft tissues and ear reconstruction. Their role in enhancing vascularization and supporting the integration of engineered tissues underscores their importance in advancing tissue engineering strategies. Future studies focusing on optimizing the use of EPCs in conjunction with other regenerative cell types will likely lead to improved outcomes in the treatment of complex wounds and reconstructive surgeries.

## Future research directions for EPCs
### Reprogramming of EPCs into induced pluripotent stem cells
Induced pluripotent stem cells (iPSCs) are a type of pluripotent stem cell that can be generated directly from adult cells. They are remarkable because they possess the ability to differentiate into almost any cell type, similar to embryonic stem cells, but without the associated ethical concerns. This capability makes iPSCs a powerful tool for regenerative medicine, disease modeling, and drug screening. The process of reprogramming somatic cells to become iPSCs involves the introduction of specific transcription factors, which effectively "reset" the cell's identity to a pluripotent state (*Chari & Mao, 2016*; *Menon et al., 2016*). The conversion of EPCs into iPSCs is intriguing as it demonstrates cellular reprogramming's flexibility and potential. EPCs, crucial for blood vessel repair, can be

reprogrammed into iPSCs, broadening their applications. This process highlights iPSCs' versatility and offers new research and treatment possibilities, especially for vascular diseases and regenerative medicine (*Orqueda, Giménez & Pereyra-Bonnet, 2016*; *Polanco, Kuang & Yoon, 2020*).

To transform somatic cells into pluripotent ones, specific transcription factors like the Yamanaka factors (Oct4, Sox2, Klf4, and c-Myc) can be used to reprogram cells such as fibroblasts and EPCs into iPSCs (*Xie et al., 2014*). This reversion to pluripotency allows cells to differentiate into various lineages, including endothelial cells, essential for vascular repair. Additionally, the microenvironment, enhanced by biomaterials and growth factors, plays a crucial role in cell fate decisions and can improve reprogramming efficiency by influencing EPCs' differentiation or reprogramming pathways (*Rahman et al., 2010*; *Farkas et al., 2020*).

Researches on deriving EPCs from hESCs and/or human iPSCs are vital for regenerative medicine and vascular biology. Effective differentiation protocols are essential. One approach uses high-capacity helper-dependent adenoviral vectors (HDAdVs) for precise gene targeting without DNA breaks, facilitating accurate gene knockout and knock-in (*Aizawa et al., 2012*). Another promising method employs zinc-finger nucleases (ZFNs) for precise genome editing, allowing for the creation of lineage-specific reporters and gene expression modification to guide stem cell differentiation. This precise editing is crucial for directing stem cells into EPCs (*Hockemeyer et al., 2009*). Furthermore, the implications of successfully reprogramming EPCs to iPSCs extend beyond basic research. This capability could lead to advancements in personalized medicine, where patient-derived cells can be used to model diseases, screen drugs, and develop tailored therapies. The potential to generate a renewable source of endothelial cells from iPSCs derived from EPCs could also address the challenges associated with cell scarcity in transplantation therapies (*Ge et al., 2018*; *Eminli et al., 2021*).

Reprogramming EPCs into iPSCs using small molecules like CHIR99021, RepSox, VPA, Forskolin, 616452, and BIX-01294 marks a significant advance in stem cell research and regenerative medicine. This approach, which also involves transcription factors and engineered microenvironments, holds potential for innovative therapies in treating vascular diseases and advancing regenerative medicine.

### *Research prospect of EPCs*

The discussion on EPCs applications and future directions is an evolving field of research that holds significant promise for advancing medical science, particularly in the areas of regenerative medicine and vascular biology. EPCs are known for their potential to contribute to the repair and regeneration of damaged blood vessels, making them a focal point in studies related to cardiovascular diseases, wound healing, and tissue engineering. The research surrounding EPCs has expanded to explore their mechanisms of action, functional characteristics, and potential clinical applications, particularly in the context of ischemic diseases and cardiovascular disorders.

The signaling pathways that regulate EPCs function are also a focus of current research. Calcium signaling, for example, has been shown to play a pivotal role in EPCs proliferation

and migration, with store-operated calcium entry being a critical mechanism driving these processes (*Moccia et al., 2014*). Additionally, the dysregulation of signaling pathways, such as the CXCR4/JAK-2 pathway, has been implicated in impairing the angiogenic capacity of EPCs, particularly in models of ischemic disease (*Cheng et al., 2019*). Understanding these pathways could lead to novel therapeutic strategies aimed at enhancing EPCs function and improving outcomes in patients with cardiovascular diseases. Moreover, the paracrine effects of EPCs are gaining recognition as a vital aspect of their therapeutic potential. EPCs secrete a variety of growth factors and cytokines that can promote tissue repair and regeneration. For instance, the secretion of VEGF and other angiogenic factors has been shown to enhance the survival and function of surrounding cells, including cardiomyocytes, in ischemic conditions (*Hong et al., 2021*; *Zhao et al., 2018*). This paracrine signaling mechanism underscores the importance of EPCs not only as direct contributors to neovascularization but also as modulators of the local microenvironment. In addition to their roles in cardiovascular repair, EPCs have been implicated in various pathological conditions, including COPD and cancer. Research has indicated that EPCs populations may be altered in these diseases, affecting their functionality and contributing to disease progression (*Bianconi et al., 2018b*). This highlights the need for further studies to elucidate the relationship between EPCs and different disease states, as well as their potential as biomarkers for disease prediction and progression.

EPCs-based therapies have gained attention for their potential in treating vascular conditions, particularly ischemic issues like myocardial injury. A study demonstrated that using superparamagnetic iron oxide (SPIO) nanoparticle-conjugated CD34 antibodies can enhance EPCs recruitment to ischemic areas, improving heart revascularization (*Sun et al., 2022*). Besides heart conditions, EPCs therapies show promise in treating diabetes-related vascular dysfunction by aiding endothelial repair and mitigating complications (*Georgescu et al., 2011*). The combined use of EPCs transplantation and simvastatin has been studied for improving angiogenesis and reducing apoptosis in ischemic conditions. In a mouse model of hindlimb ischemia, this combination significantly improved blood flow and capillary density, while decreasing muscle cell apoptosis. This indicates that EPCs paired with other therapies can boost vascular repair and regeneration (*Hu et al., 2008*).

In conclusion, the research prospect of EPCs is vast and multifaceted, encompassing their roles in vascular biology, mechanisms of action, and therapeutic applications. Continued exploration of EPCs biology, including their signaling pathways, paracrine functions, and interactions with other cell types, will be essential for harnessing their full potential in regenerative medicine and improving cardiovascular health.

## CONCLUSION

Research on EPCs is gaining attention, particularly in relation to angiogenesis and endothelial lesions. However, challenges like unclear differentiation, low mobilization, and clinical translation difficulties persist. Future efforts should integrate omics technologies and standardized training to advance from basic research to precision treatment. This review aims to provide a comprehensive understanding of EPCs and highlight the challenges in their research.

### Funding

This work was supported by Zhejiang Province Medical and Health Science and Technology Plan (2023KY1043); Zhejiang Provincial Natural Science Foundation of China under Grant (LTGY24H150001). The funders had no role in study design, data collection and analysis, decision to publish, or preparation of the manuscript.

### Grant Disclosures

The following grant information was disclosed by the authors:
Zhejiang Province Medical and Health Science and Technology Plan: 2023KY1043.
Zhejiang Provincial Natural Science Foundation of China: LTGY24H150001.

### Competing Interests

The authors declare there are no competing interests.

### Author Contributions

- Gongjie Ye conceived and designed the experiments, analyzed the data, prepared figures and/or tables, authored or reviewed drafts of the article, and approved the final draft.
- Yongfei Song conceived and designed the experiments, performed the experiments, prepared figures and/or tables, and approved the final draft.
- Yiru Weng performed the experiments, prepared figures and/or tables, and approved the final draft.
- Jiangfang Lian analyzed the data, authored or reviewed drafts of the article, and approved the final draft.
- Jianqing Zhou performed the experiments, authored or reviewed drafts of the article, and approved the final draft.
- Zhouzhou Dong conceived and designed the experiments, analyzed the data, authored or reviewed drafts of the article, and approved the final draft.

### Data Availability

 This is a literature review.

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
