# Peer review of "Unlocking the potential of endothelial progenitor cells: a comprehensive review of definitions, applications, and future directions"

_PeerJ, doi:10.7717/peerj.20128_

## Round 0.1 · original submission · Major Revisions

We have received three detailed reviews of your manuscript.

·

Basic reporting

The authors of the manuscript titled ‘Unlocking the potential of endothelial progenitor cells: A comprehensive review of definitions, applications, and future directions’ provides a comprehensive review of endothelial progenitor cells (EPCs), covering their definitions, classification, extraction, cultivation, and clinical relevance. However, there are significant issues that make it unsuitable for publication in PeerJ in its current state.
The literature review offers a comprehensive overview of endothelial progenitor cells (EPCs), detailing their pivotal role in regenerative medicine and cardiovascular therapy. The structure of the review appears clear, systematically introducing key aspects such as the definition, classification, extraction methods, and sources of EPCs. The inclusion of strategies for optimizing EPC cultivation techniques and the potential for reprogramming them into induced pluripotent stem cells (iPSCs) reflects an advanced understanding of the field.
However, there are significant issues with clarity, consistency, and presentation. The manuscript is mostly descriptive, summarizing previous research without sufficiently analyzing conflicting results, and limitations. One of the major weaknesses of this manuscript is the lack of background information on EPC research, including its historical context and initial discoveries. Endothelial cells (ECs) exist in various subtypes, and there are ongoing debates regarding their classification and functional distinctions. However, the authors primarily focus on reviewing their own previous studies, rather than presenting a balanced discussion of the broader field. A more effective approach would be to introduce the different types of ECs in the background section, highlight the major debates in the field, and then focus the review specifically on EPCs. This structure would greatly enhance the clarity and accessibility of the manuscript for readers.

Experimental design

The Search Methodology section outlines a systematic approach to identify the necessary literature for the review. However, improvements are needed.
The described meta-analysis methodology should be presented as a flowchart in the main text. This will enhance clarity and allow readers to follow the research selection process more easily. The manuscript extensively discusses EPC mechanisms of action, vasculogenesis, angiogenesis, and signaling pathways. However, these concepts are not supported by visual representations. To improve clarity and engagement, diagrams and schematic illustrations should be incorporated to visually depict these processes.
The Search Methodology section is missing the URLs of the online databases, which should be included following the PeerJ author guidelines (https://peerj.com/about/author-instructions/#reference-format).
The flow of topics in the paper is unnatural. To enhance the reader's understanding and ensure a more logical progression of topics, I suggest the following revision to the structure of the manuscript:
Search Methodology
1. Definition and Classification of Endothelial Progenitor Cells (EPCs)
(1) EPCs vs. Late EPCs
2. Sources and Extraction Methods of EPCs
(1) Various Sources of EPCs
(2) Extraction Methods
(3) Optimization of EPC Cultivation Techniques
3. EPCs in Regenerative Medicine and Disease Treatment
(1) EPCs and Cardiovascular Disease (CVD)
(2) EPCs and Septic Shock
(3) EPCs in Rheumatic Immunity and Connective Tissue Diseases (CTDs)
(4) EPCs in Skin and Soft Tissue Regeneration
4. Future Research Directions for EPCs
(1) Reverse Transformation of EPCs into Induced Pluripotent Stem Cells (iPSCs)
(2) Research prospect of EPCs

Validity of the findings

A more detailed breakdown of EPC subtypes and their differing characteristics might offer the reader a more nuanced understanding.
More detail on how EPCs specifically interact with the pathophysiology of each disease (e.g., cardiovascular diseases, septic shock, or autoimmune conditions) would be useful. Additionally, including recent studies or clinical trials could make the review more up-to-date and relevant.
The vague discussion on EPC applications and future directions suggests a lack of deep engagement with the latest research trends.
The references cited by the authors are relatively outdated. To strengthen the manuscript and provide more current insights, it would be beneficial to incorporate recent studies that reflect the latest research trends.

Additional comments

The manuscript needs major revisions before it can be considered for publication. The literature review methodology must be explicitly stated, and the analysis should be more critical. The writing should be proofread for grammar, consistency, and clarity. More figures and tables should be incorporated to improve readability and support the findings.
There is significant redundancy throughout the manuscript. Similar discussions and conclusions appear repeatedly in different subsections, disrupting the overall coherence and flow of the review. A more structured organization is required to ensure that key points are presented logically and without unnecessary repetition.
To enhance the scientific quality and readability of the review, the authors should consider visualizing key concepts through flowcharts, signaling pathways, and summary tables. Incorporating such graphical elements would elevate the level of the review and significantly improve the reader’s comprehension.
Revision Suggestions:
Line 24: Nonetheless, numerous challenges and questions persist regarding the practical research and application of EPCs.
→ Nonetheless, numerous challenges and questions persist regarding the translational research and practical application of EPCs.
Line 43: The described meta-analysis methodology should be presented as a flowchart in the main text. This will enhance clarity and allow readers to follow the research selection process more easily.
Line 36: Endothelial progenitor cells (EPCs) play a pivotal role in regenerative medicine and cardiovascular therapy, primarily due to their involvement in neovascularization and vascular repair.
→ Endothelial progenitor cells (EPCs) have shown promise in regenerative medicine and cardiovascular therapy, primarily due to their involvement in neovascularization and vascular repair. However, their clinical efficacy remains under investigation.
Line 54: IEEEXplore → IEEE Xplore® (https://ieeexplore.ieee.org)
Line 62: A clear explanation of "conflict classification" and "conflict detection method" would be helpful to readers.
Line 72: Illustrating the characteristics of vasculogenesis and angiogenesis in an image would help readers understand the concepts more clearly.
Line 74: The nomenclature surrounding EPCs has been a subject of debate, with calls for a more precise definition that includes detailed immunophenotyping and functional assays to differentiate between various endothelial progenitor subtypes.
→ The definition of EPCs remains debated, with researchers advocating for immunophenotyping and functional assays to distinguish different EPC subtypes.
Line 86 and 304: Our research → Previous studies
Line 127: Comparing the similarities and differences between early EPCs and late EPCs in a table would help clarify the concepts for readers.
Line 180: Please describe the methods for obtaining EPCs from hESCs and/or hiPSCs.
Line 155: Fluorescence-activated cell sorting (FACS) is also a representative method, so it would be appropriate to include it.
Line 211: Please describe the conditions and considerations for the clinical application of EPCs as cell therapies.
Line 229: ECFC → Please clarify whether EPCs and ECFCs refer to the same or different cell populations.
Line 247: The term "reverse transformation" is less appropriate, and "reprogramming" would be more suitable.
Line 249: Recent studies → Please add a reference.
Line 264: Please suggest specific small molecules.
Line 267: Please provide examples of specific inhibitors or activators.
Line 269 – 277: The section discusses general advantages of using hPSCs for cell therapy, which are not unique to EPCs.
Line 286: circulating EPC → It would be beneficial to mention the definition and characteristics of circulating EPCs in the previous paragraph.
Line 309: endothelial progenitor cells → EPCs
Line 319: high mortality rates → Please provide specific rate (%) for the high mortality rates and their reference.
Line 407 – 411: It should be included in the introduction.
Line 445 – 446: Please mention the specific shortcomings and uncertainties.
Line 728: Please fill in the first empty cell of the table with the category.
Etc.:
- Please use superscripts for positive (+) markers by replacing regular symbols with superscript symbols (⁺). (CD34+CD133+KDR+ → CD34⁺CD133⁺KDR⁺)
- Please use periods (.) when listing subtopics 1 – 11.
- Could you kindly clarify why the most representative EPC paper, Asahara 1997 (https://www.science.org/doi/10.1126/science.275.5302.964), was not mentioned in the review? The reason for its omission is not immediately clear.
- Please explain the methods for purifying high-purity EPCs.
- Please explain the next steps to improve the clinical application of EPC-based cell therapies.
- Overall, the flow between paragraphs is somewhat incoherent and could benefit from smoother transitions, which would create a more natural progression of ideas. Additionally, repetitive expressions and sentence structures should be streamlined to improve clarity, consistency, and overall flow.

·

Basic reporting

The topic is interesting and has the potential to grab the attention of EPC research peers as a mild introduction to EPCs and their potential.
1. The introduction is merely the same as that of the abstract. I suggest that authors should cite/enlist the importance of EPCs in research to grasp the attention of the readers.
2. Line 36 (This review is specifically aimed at researchers focused on EPCs) should be excluded, as it presents a potentially restrictive or exclusive nature of the information provided in this study. Readers without prior knowledge of EPCs should be interested in reading the article.
3. Line 74 ( CD34 CD133 KDR triple-positive EPCs) should be corrected to CD34+CD133+KDR+.
4. Figure 1: Line 670-672: A and B image, please include a missing description in the figure legend.
5. Line 99, UEA should be defined. With each new paragraph, an abbreviation (uncommon) should be defined for ease of understanding.
6. The ‘extraction method of EPCs’ should be corrected to the ‘extraction methods of EPCs’.
7. Line 226-230: should include the essential (short/crisp) information of what iPSCs are and why the reverse transformation of EPCs into iPSCs is so fascinating.
8. Line 244: define Wnt or BMP signalling cascades.

Experimental design

Logically, ‘3. Extraction methods of EPCs should come after ‘4. The source of EPCs. This also resolves the confusion of ‘how?’ from lines 150-151 [peripheral blood mononuclear cells (PBMCs), which can then be further processed to obtain EPCs].

Validity of the findings

More depth of discussion regarding the known research discussed in the study would benefit the article.

In conclusion, stating the shortcomings and uncertainties (Line 402-403) will be helpful to remove the bias and provide a thorough insight into the review.

Additional comments

The authors should consider generating a schematic to drastically improve the impact, catch the attention, and improve the understanding of the readers.
Example – (1) The role of EPCs in various diseases such as CVD, Septic shock, immunity, CTDs, etc. (2) The role of EPCs in tissue regeneration.

·

Basic reporting

The manuscript occasionally uses “EPCs” in a confusing manner, sometimes as a singular or plural noun. Standardizing the terminology (e.g., “EPC populations” or “EPC function”) would improve clarity and professionalism.

Experimental design

no comment

Validity of the findings

no comment

Additional comments

1. In lines 72-78, the paragraph repeats information unnecessarily: both sentences mention clonogenicity and angiogenic potential as key features without adding new insight. The authors suggest triple-positive cells as pro-angiogenic.
2. In - Divergence in the definition of endothelial progenitor cells (EPCs) - The text relies on surface markers (CD34+/CD133+/KDR+) to define EPCs, but does not address ongoing controversies about their true identity. Recent studies argue that these markers are shared by hematopoietic progenitors, risking misclassification. A stricter functional validation (e.g., tube formation assays in vitro or revascularization in vivo) is needed to distinguish true EPCs from circulating endothelial cells or monocytes. Please clarify to be clearer. Also, ac-LDL/UEA-1 binding is presented as definitive for EPC identification, but macrophages and mature endothelial cells also exhibit these traits. Dual-staining with lineage-negative markers (e.g., CD14−/CD45−) or concurrent functional tests (e.g., migratory capacity) would improve specificity.
3. In the early vs late EPCs section, the assertion that late EPCs have "superior proliferation" conflicts with data showing CAD patients exhibit more late EPC colonies but reduced CD31/KDR expression—a sign of immaturity. Similarly, early EPCs from hypertensive patients show impaired repair capacity despite normal numbers, highlighting the proliferative potential ≠ and functional efficacy.
4. The text assumes early EPCs mature into late EPCs, but evidence suggests they arise from distinct lineages: early EPCs derive from monocytes/macrophages, while late EPCs (ECFCs) may originate from vascular endothelia.
5. In early vs late EPCs - UCB-derived EPCs exhibit significantly higher proliferative capacity and delayed senescence compared to adult peripheral blood or bone marrow sources, a critical point omitted in the original analysis. Please include this critical point in the text.
6. In the cultivation of EPCs section, the section does not consistently specify whether findings apply to all EPC subtypes (e.g., ECFCs vs. early EPCs), potentially leading to confusion about the generalizability of the techniques described.
7. Although clinical potential is mentioned, there is little discussion of practical challenges (e.g., scalability, cost, regulatory issues) that may impact the adoption of these optimized cultivation techniques in real-world settings
8. The reprogramming section relies heavily on general reprogramming strategies (e.g., Yamanaka factors) but provides little direct evidence or data specific to the reprogramming of EPCs compared to other cell types. Also, “reversing EPCs into iPSCs” is not standard terminology (“reprogramming EPCs to iPSCs” would be more precise).
9. “back to a pluripotent state” is redundant; “to a pluripotent state” is sufficient.
10. EPCs and CVD - The text alternates between “EPCs levels” and “EPC levels,” which should be standardized for clarity and grammatical correctness.
11. There is no discussion of limitations in the cited studies, such as variability in EPC measurement methods, patient heterogeneity, or the challenges of translating preclinical findings to the clinic
12. For prospects of EPCs, what about the current obstacles, such as variability in EPC isolation, immune responses, and the need for standardized protocols?

---

## Round 0.2 · Minor Revisions

Please address the remaining issues pointed out by Reviewer 1 and amend the manuscript accordingly.

·

Basic reporting

This reviewer has carefully reviewed the revised manuscript entitled, "Unlocking the potential of endothelial progenitor cells: A comprehensive review of definitions, applications, and future directions". This reviewer appreciates the authors’ efforts in addressing the comments in the previous review.
The revised manuscript has shown improvement by incorporating the suggestions provided by the peer reviewers. Notably, the Introduction presents a more valuable background and context for the study. The overall structure of the manuscript has been refined, making it easier for readers to understand and improving readability. Additionally, the methodology section has been further clarified and detailed, and relevant literature has been cited more comprehensively and appropriately.

However, this reviewer has found that there remain a few issues that would benefit from further minor revisions to enhance the manuscript for publication.

Experimental design

The added flowchart illustrating the methodology and the schematic diagram visually depicting the mechanism will likely enhance readers’ understanding. The reorganization of the manuscript’s structure has also contributed to a more logical progression and improved readability. Overall, the authors’ revisions have substantially strengthened the manuscript.

Validity of the findings

The authors have added analyses of EPC subtypes and their roles in disease pathophysiology, along with their clinical correlations. Furthermore, by including recent studies and clinical trials, the review’s relevance and overall quality have been significantly enhanced.

Additional comments

Minor comments:
1. Line 30: The first letter 'e' of 'endothelial' appeared in bold font, which should be corrected.
2. Line 239: A space error occurred after 'potential'.
3. Line 406: Please remove the superscripted period.
4. Figure 1A: Adjusting the spacing and size of the elements would improve the clarity of the message.
5. Figure 1B: EPC and EC are depicted in very similar colors, which may confuse. Using more distinctly different colors would improve clarity.
6. Please double-check the source attribution for the added illustration image, as it may be required to show it. For example, according to BioRender’s guidelines, the phrase "Created with BioRender.com" must be clearly indicated.
7. Using abbreviations for frequently repeated terms could enhance readability for the readers. For example, "Umbilical Cord Blood" can be abbreviated as "UCB," and "Peripheral Blood" as "PB."
8. The surface marker VEGFR-2⁺ among early EPCs in Table 2 is incorrectly formatted with a superscript ‘2’ and needs to be corrected.
9. Line 888, Table 3: Reference number 76 is incorrect. It should be corrected to 67.

·

Basic reporting

Good job on improving the manuscript significantly. Good luck to the authors for their future endeavours!

Experimental design

Meets the standards

Validity of the findings

-

---

## Round 0.3 · accepted · Accept

Since all remaining concerns of the reviewers were adequately addressed, the revised manuscript is acceptable now.